

**Domestic water consumption pattern by urban households**
Amarasingam Narmilan[1], Narmilan Puvanitha[2], Gnanachelvam Niroash[3], Muthucumaran
Sugirtharan[4], and Ratnarajah Vasssanthini[5]
[1, 3,5]Department of Biosystems Technology, Faculty of Technology, South Eastern University
of Sri Lanka
[2]Department of Agriculture, Hardy, Sri Lanka Institute of Advanced Technological Education.
[4]Department of Agricultural Engineering, Faculty of Agriculture, Eastern University, Sri
Lanka
*Corresponding Author Email: *narmilan@seu.ac.lk
**ABSTRACT**
Water has been recognized as one of the most significant natural resources and crucial for health and wealth.
The increased demand for water has imposed pressure on the water supply system, which has led to
environmental problems such as over-exploitation of water resources and breaks in the balance of the
ecosystem. Determining the behavior of domestic water consumers can facilitate a more proactive approach
to water demand management, and serves as the foundation for the development of any intervention
strategies that seek to bring about sustained and substantial reductions in domestic water consumption. This
study tried to investigate household water consumption patterns and management practices along with
comparing the effectiveness of different water management measures on reducing the water deficit of the
district. The primary data was collected through a questionnaire survey from 75 households belonging to the
urban area in Batticaloa District in Manmunai Pattu, Sri Lanka. The data were analyzed both quantitatively and
qualitatively. The findings show that people with higher incomes in urban areas are using more water than
people with lower incomes. The water usage depends on the living standards, family size, age, and education
level of household members and the number of taps present in the household. It is believed that the results of
the study would be beneficial for domestic water consumption in urban Batticaloa.
**KEYWORDS**: *Batticaloa; Efficiency; Household; Water consumption; Water deficit*

**INTRODUCTION**
Water has been played a crucial role in the location, function, and growth of communities. Water is
essential to life and it serves as the base for the social and economic development of any country in
the world (Omvir and Sushila, 2013). The United Nations has projected world population would
increase by an additional two billion ($2 \times 10^9$) people by the year 2030 (Postel, 2000). The World
Health Organization (WHO) defined domestic water as the water used for all domestic purposes
including drinking, bathing, and food preparation. Domestic water consumption is a significant
component of the total water use and it varies according to the living standards of the consumers in
urban and rural areas (Mohammed and Sanaullah, 2017). Water is used for various indoor purposes
among which are bathing, washing clothes, drinking, flushing the toilets, washing plates, washing
fruits and vegetables, brushing teeth, cooking, performing ablution, and shaving (Olasumbo, 2006).
Providing adequate and improved drinking water is an increasingly significant albeit a daunting
challenge for authorities, development agencies, and water sector organizations, more especially in
countries with rapidly growing populations. Improved drinking water refers to water sourced from a
tap located within premises or yard/plot, a public standpipe, a tube well, a protected dug well or
spring, and rainfall (UNICEF/WHO, 2015).
Population growth, expansion of business activity, urban development, water pollution, climate
change, and drought have contributed to increased water scarcity in many parts of the world. It is
estimated that a fifth of the world's population live in areas of physical water scarcity, where there is
not enough water to meet all demands. One-third of the world's population does not have access to
clean drinking water. Further one-fourth of the world's people live in areas of economic water
scarcity, where poor management makes it impossible for authorities to satisfy the demand for
water (Molden, 2007). The household water consumption is determined by quite a few factors, such



as climate, seasonality, socioeconomic characteristics, and socio-demographics. In this study, only
the socio-demographic factors are taken into account. The majority of research projects have
focused on highlighting the current water shortage and the increased use by the residential sector.
However, a lack of studies on household water consumption is observed when meeting household
water demand is one of the main goals of various policy interventions and programme guidelines on
drought mitigation or domestic water management strategies. The present study aims at analyzing
the impacts of household socio-economic conditions on various aspects of domestic water
consumption in urban Batticaloa in Manmunai Pattu, Sri Lanka.

**METHODOLOGY**

A survey was conducted on household water consumption in urban Batticaloa area. This survey
includes the development and distribution of a questionnaire to the households of urban Batticaloa.
A Simple random sampling technique was followed to select households such that each household
has an equal probability of being included in the study. Besides, more than half of the respondent
households do not engage in water conservation at their households at present due to continuous
access to water through their water source.
*Flow rate experiment*
The results of the semi-structured interview showed that the sales assistants in water appliances
shops were not sure about the flow rate of taps and showerheads. They identified some water-
efficient products but were not sure how much water could be saved. Product instruction only
showed the size and features of the product, not including the flow rate. The varying flow rates of
different appliances could affect water consumption in different households. So, the flow rate is an
important indicator to understand the amount of water use at home. From the literature review, it
was found that the flow rate (tap and showerhead) could be measured through a simple
experiment. The test procedure was based on the Green Venture website: how to conduct a flow
rate test, 2007(Green Venture, 2007). The test instruments included a stopwatch (Mobile phone), a
container with measurements on the side, the maximum measurement being 1.5 litres, and a
calculator. The main procedures were as follows:
1) The empty container was placed under a tap or showerhead; the tap or the showerhead was
turned on to its highest flow rate. The stopwatch was started at the same time. When the water
reaches 1 litre, the watch was stopped and the time was recorded.
2) The flow rate was calculated. For example, to fill one litre container takes 5.8seconds, 5.8 sec= 0.1
min, the flow rate = 1 litre/ 0.1 minute= 10 litre / minute
3) This procedure was repeated twice for each test and the average number was used.









**RESULTS AND DISCUSSION**
*1. Demographic composition*
Table 1: Demographic composition

| Age of the household head (years) | Number | Percentage | Education | Number | Percentage |
|---|---|---|---|---|---|
| Below 25 | 0 | 0 | Primary | 0 | 0 |
| 25-35 | 8 | 10.7 | Intermediate | 15 | 20.0 |
| 36-45 | 16 | 21.3 | Advanced | 36 | 48.0 |
| 46 -55 | 21 | 28.0 | Higher | 22 | 29.3 |
| 56-65 | 23 | 30.7 | None | 2 | 2.7 |
| Above 66 | 7 | 9.3 | **Total** | **75** | **100.0** |
| **Total** | **75** | **100.0** | | | |
| Ownership of the House | | | Living standard of the family | | |
| Own | 64 | 85.3 | Poor | 2 | 2.7 |
| Rented | 11 | 14.7 | Medium | 59 | 78.7 |
| **Total** | **75** | **100** | Rich | 14 | 18.7 |
| | | | **Total** | **75** | **100.0** |
| Occupation of Household head | | | Average Monthly Income of Household | | |
| Government | 29 | 38.7 | Below10,000 Rs | 0 | 0 |
| Private/NGO | 11 | 14.7 | 10,001-15,000 Rs | 2 | 2.7 |
| Business | 7 | 9.3 | 15,001-20,000 Rs | 2 | 2.7 |
| Farmer | 4 | 5.3 | 20,001-25,000 Rs | 6 | 8.0 |
| Day-wage labour | 4 | 5.3 | 25,001-30,000 Rs | 12 | 16.0 |
| Others | 20 | 26.7 | 30,001-40,000 Rs | 11 | 14.7 |
| Total | **75** | **100.0** | 40,001-50,000 Rs | 13 | 17.3 |
| | | | Above 50,000 Rs | 29 | 38.7 |
| | | | **Total** | **75** | **100.0** |
| Family size | | | | | |
| 2 | 2 | 2.7 | | | |
| 3 | 26 | 34.7 | | | |
| 4 | 27 | 36.0 | | | |
| 5 | 11 | 14.7 | | | |
| 6 | 6 | 8.0 | | | |
| 7 | 3 | 4.0 | | | |
| **Total** | **75** | **100** | | | |


Different Statistical analyses were carried out with the assistance of IBM SPSS Software (Version
25.0) and the data were presented. Simple descriptive measures, analysis for variance, post hoc
tests, and multivariate regression analysis were applied. The principal component analysis was used
to assess the socio-economic status of households based on the assets they hold. Before any
parametric statistical analysis, data were assessed for normality. The demographic composition of



the sample households/Social status of farmers in the survey community is shown in Table 1. The
age distribution and the education level of the heads of these households are shown in Table 1.
Around 30.7% of households' heads are aged between 56 to 65 and 28% are aged from 46 to 55
years while those who in 36 -45 age accounted for 16% of the total respondents. With regards to the
household heads whose age between 25 -35 years and below 66 years were almost similar by having
8% and 7% respectively. However, there were no household heads observed below 25 aged groups.
The survey showed that around half of the respondents (48%) have completed their advanced level
of education while those who have received their higher education and intermediate level of
education are 22% and 15% respectively. However, only 2% of them were uneducated and there are
no individuals who attained only primary education. The result in Table 1 shows that 85.3% of
household heads have their own house while 14.7% of respondents reside in rented houses. In
terms of living standards of the respondent's family, it was observed that a higher percent (73.70%)
of the family whose living standard is medium followed by rich families (18.7%) while the poor were
accounted for 2.7%. The number of household size is one of the basic demographic characteristics of
a household. Distribution of respondents according to household size shows that the majority (36 %)
of the families had 3 to 6 members in their houses while 14% of them had 5 members and those
who have the members of 6, 3, and 2 in 8%, 4%, and 2% respectively. According to the survey, the
occupations of family heads found to be involved in the government sector (38.7%), other kinds of
jobs (20%), private or NGOs (14.7%), and the rest of them were engaged in business (9%), farming
(5.3%) and daily labour work (5.3%).

*2. Age of household members*
Water usage is also affected by age of household members. The water usage behaviors can be quite
different among different ages of household members. Households with children could be expected
to use more water. Youngsters might use water less carefully, e.g. taking more showers, doing more
frequent laundering, while retired people might be much thriftier (Nauges and Thomas, 2000).  Elder
people use less water than younger people. Nauges and Thomas (2000) support this and observe
that communities with more seniors have lower water consumption, and similar results have been
found by (Martínez-Espiñeira 2002, Martins and Adelino 2007, Musolesi and Nosvelli 2007). But
Schleich and Hillenbrand (2009) found the opposite, that the elder people use more water because
retired people spend more time at home and gardening. After all, children use less water for
washing and hygiene than adults, or because health reasons may force older people to use the
bathroom more frequently.
*3. Living standards*

Table 2: Correlation between living standards – total usage

|  |  | Living standards | Total usage |
|---|---|---|---|
| Living standards | Pearson Correlation | 1 | .825** |
|  | Sig. (2-tailed) |  | .000 |
|  | N | 75 | 75 |
| Total usage | Pearson Correlation | .825** | 1 |
|  | Sig. (2-tailed) | .000 |  |
|  | N | 75 | 75 |

**. Correlation is significant at the 0.01 level (2-tailed).


Total domestic water consumption is positively correlated with living standards as *p*<0.01 (Table 2).
This was supported by Syme et al. and Loh and Coghlan.  This result is attributed to the use of
modern appliances and a lack of knowledge of elders. People in developing countries spend more
money on items that consume more water such as dishwashers, washing machines, flushing toilets,
and showers.  People also tend to eat more meat as living standards increase, which also needs

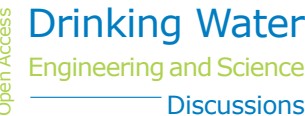

more water in its production. A variable that has a positive effect on household water consumption
is the number of people in a residence (Hanke and Maré 1982). Total water usage of the study
population was 12732.5 liters and Per capita, water usage was 169.8 liters.

*4. Income level*
The correlation between water consumption and income level of the survey community is shown in
Table 3.
Table 3: Correlation between water consumption and income level

|  |  | Income | Total usage |
|---|---|---|---|
| Income | Pearson Correlation | 1 | .968[**] |
|  | Sig. (2-tailed) |  | .000 |
|  | N | 75 | 75 |
| Total usage | Pearson Correlation | .968[**] | 1 |
|  | Sig. (2-tailed) | .000 |  |
|  | N | 75 | 75 |

**. Correlation is significant at the 0.01 level (2-tailed).


It is shown that the total domestic water consumption is positively correlated with income level
($p<0.01$). High water consumption may due to the high living standard of the survey community
(Table 3), as a high level of income is associated with high living standards. This may mean a higher
number of water-consuming appliances and a higher probability of high-water usage for watering
large garden areas. This was supported by Guhathakurta and Gober, (2007) who indicate that
income rises result in a corresponding increase in water consumption. Dalhuisen, 2003, stated that
though the water consumption is increased with income, it is not a proportional increase. Usage of
western-style bathtubs, dishwashers, and washing machines in high-income households also
attribute to high-water consumption. The literature by Kenney, 2008 has also reported higher water
consumption per capita for higher-income homes.
*5. Education level*
The correlation between water consumption and education level of the survey community is shown
in Table 4.
Table 4: Correlation between water consumption and education level

|  |  | Education | Total usage |
|---|---|---|---|
| Education | Pearson Correlation | 1 | -.873[**] |
|  | Sig. (2-tailed) |  | .000 |
|  | N | 75 | 75 |
| Total usage | Pearson Correlation | -.873[**] | 1 |
|  | Sig. (2-tailed) | .000 |  |
|  | N | 75 | 75 |

**. Correlation is significant at the 0.01 level (2-tailed).


The education level also influences the water consumption in a household. It is shown that the total
domestic water consumption is negatively correlated with education level as $p<0.01$ (Table 4).
Educated people are more conscious about the increasing water scarcity and they literate their
younger generation to use the water resources efficiently. It has been shown in (Millock and Nauges,
2010) that the education level is positively correlated with lower water consumption and higher



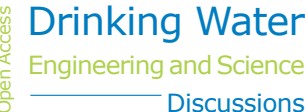

water conservation behaviors which would cut down the household total water consumption.
Educational campaigns teach easy ways to conserve water and increase feelings of self-efficacy.
Targeted educational campaigns about environmental conservation behaviors aimed at elementary
students in the US are effective in increasing those behaviors within their households (Woollam et
al, 2006). Keshavarzi et al, 2006 reported that the low level of education of elders regarding
environmental matters leads them to consume more water than do younger people. But in contrast,
Collins et al, 2003 stated that older people tend to use less water because of traditional practices of
water usage (washing hands, showering, and sharing water among family members) and their
unfamiliarity with water appliances.

*6. Number of taps*
Table 5: Correlation between the number of taps and total usage

| | | Number of taps | Total usage |
|---|---|---|---|
| Number of taps | Pearson Correlation | 1 | .951[**] |
| | Sig. (2-tailed) | | .000 |
| | N | 75 | 75 |
| Total usage | Pearson Correlation | .951[**] | 1 |
| | Sig. (2-tailed) | .000 | |
| | N | 75 | 75 |

**. Correlation is significant at the 0.01 level (2-tailed).


The number of taps also influences the water consumption in a household. Table 5 shows that the
total domestic water consumption is positively correlated with the number of taps as $p<0.01$. It is
proved from the results that there was a great impact on water consumption due to the increased
number of taps. Also, the increase in water consumption could be attributed to the pipe diameter
and water flow rate (Englart and Jedlikowski, 2019).
*7. Household size*
Table 6: Correlation between family size and total usage

| | | Family size | Total usage |
|---|---|---|---|
| Family size | Pearson Correlation | 1 | .950[**] |
| | Sig. (2-tailed) | | .000 |
| | N | 75 | 75 |
| Total usage | Pearson Correlation | .950[**] | 1 |
| | Sig. (2-tailed) | .000 | |
| | N | 75 | 75 |

**. Correlation is significant at the 0.01 level (2-tailed).

Table 6 shows that the total domestic water consumption is positively correlated with household
size as $p<0.01$. The number of household members affects the amount of water used in a house
(Gaudin, 2006). Households with more family members used larger quantities of water. Arbus, et al
(2004) found that water consumption increases with the household size, though it is not a
proportional increase. However, household size was found to be an insignificant factor in water
usage at the domestic level (Guhathakurta and Gober, 2007). A household of a large size normally
uses more appliances with greater frequency, resulting in more water usage than a small size
household. Numerous studies have shown a strong correlation between the age of household head
and net family size and water consumption (Arouna and Dabbert, 2010; Syme et al, 2004).
*8. Water supply*

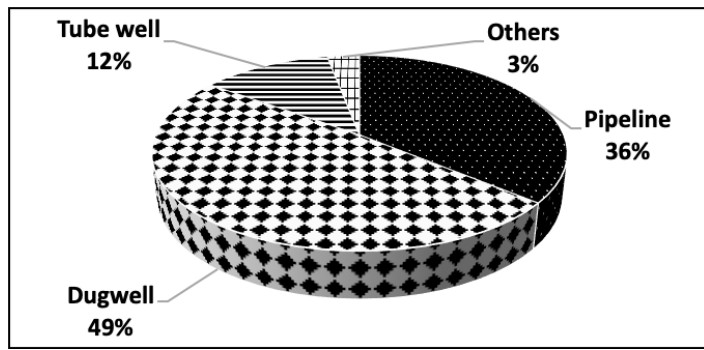


205                      Figure 1: sources of the water supply of the households

The chart (Figure 1) illustrates the different sources of the water supply of the households. It was
clear that around half of the proportion of the households (49%) receive the pipeline water followed
by tube well usage to a level of 36% while those who use water from dug well accounted for 12%.
The lowest amount (3%) of respondents got water from other sources like lakes, rivers, and ponds. A
similar result was reported by Tadesse et al. (2013) and Mahama et al. (2014). The choice of water
source is strongly influenced by several household characteristics. Local households seem to have
adopted different practices for accessing alternative water sources rather than dug well alone to
meet their diverse needs.  Most households are dependent on private wells. But water sources and
their uses changed significantly between the wet and dry seasons (Elliott et al, 2017). The most
common household water sources were taps and well (Casanova et al, 2012).
*9. Drinking water*

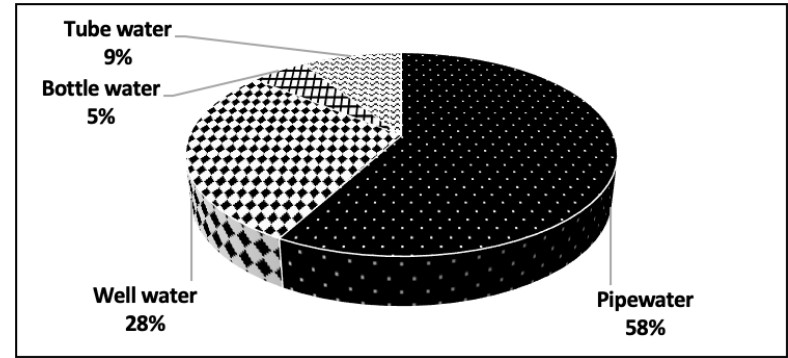


225                      Figure 2: sources of drinking water of the households

The figure summarizes the percentage use of drinking water from a different source of water supply.
Overall, the highest amount (58%) of drinking water was collected using the pipeline. Drinking water
consumption from well water accounts 28% of the total population while the tube well water and
bottled water were the lowest quantity of water which is utilized for drinking purposes among the
households for 9% and 5%. Piped water supply was the most common drinking-water source in
urban areas. This parallels the Nketiah-Amponsah et al. (2009) observed that access to a piped
drinking water source is higher compared to other types of drinking water sources. Bottled water
consumption is low due to the high price. Results of a study by Vásquez, 2017 indicated that bottled



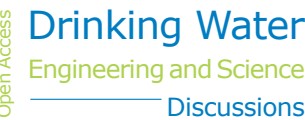

water consumption was positively associated with health risk perceptions, household income, and
education and market access. Household size negatively impacted the likelihood of consuming
bottled water.

*10. Family practice adopted in the preparation of drinking water*

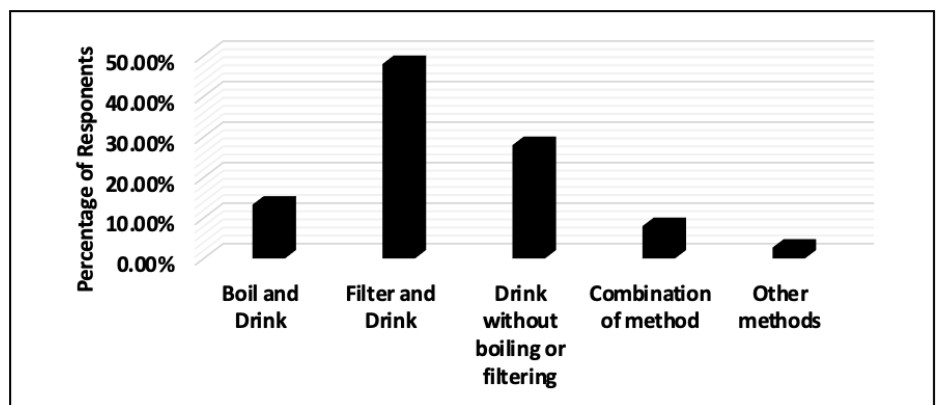

Figure 3: Family practice adopted in the preparation of drinking water
Figure 3 shows the family practice adopted in the preparation of drinking water. Most of the
respondents (48%) were practicing filter and drinking methods but 28% of the families were adopted
to drinking the water without boiling or filtering. In terms of the boiling and drinking method, only
about 13% of families were using this method. However, only about 8% of respondents were using
the combination method and 2.7% were using other methods when preparing the drinking water.
Boiling and filtering are the most common methods used in households for purifying water. Clasen
et al, 2008 stated that boiling is a relatively expensive method, and Wolf et al, 2014 stated that
filtering by cloth is an ineffective method. Gilman and Skillicorn, 1985 stated that the cost of boiling
may be expensive for many low-income populations. Francis et al, 2015 observed the frequency of
filtering water for children is higher than adults.  However, studies have shown that, although
necessary and potentially having a positive health impact, households do not regularly use HWT
(Brown and Clasen, 2012). Filtering was more common among user households than any form of
treatment (Casanova et al, 2012).

*11. Irrigation*

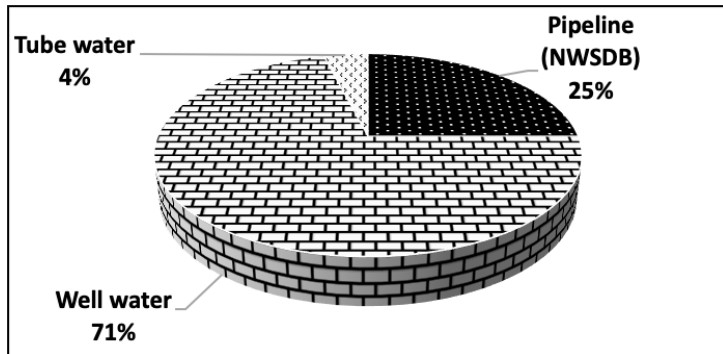



Figure 4: sources of irrigation of the households
It was clear that the highest percentage (71%) of water from well water has been used for irrigation
purposes among the households while the least amount of water for irrigation has been drawn up
using tube well. However, 25% of the water was collected from well water.

*12. Water-related appliances in the home*

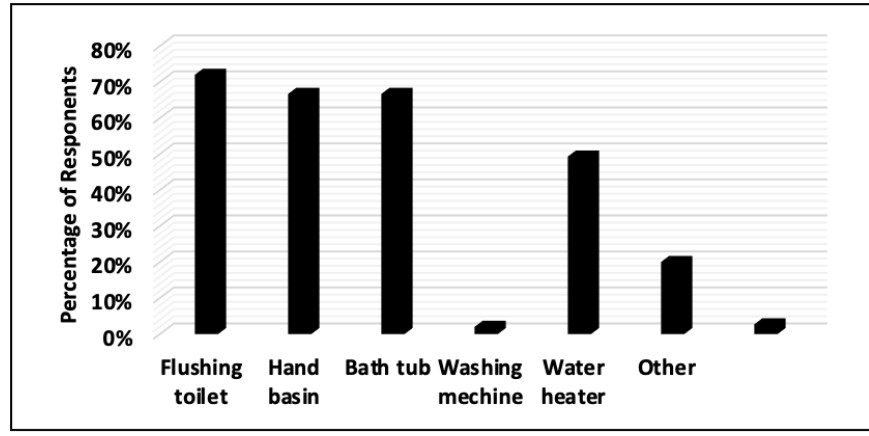


Figure 5: usage of water-related appliances in the home
The chart illustrates the patterns of water use by households. It was clear that the highest amount
(72%) water has been used for showers and baths for daily use by households while 66.7% of total
water of household is used in toilet flushing and personal hygiene, especially for hand washing.
Nearly half of the proportion of water is utilized for washing machines. It was also found that small
quantities needed for water heaters, bathtub and other needs using 20%, 12%, and 2.7%
respectively. Literature by Beal and Stewart argues that high volumes of water are consumed by
teenagers for showers. Shaban and Sharma, (2007) found that bathing, flushing, clothes washing,
and utensil washing accounting for much higher water use in households. Modern changes in
lifestyle all potentially contributing to the increase in water use for bathing and showering (Bello-
Dambatta, 2014). Also, en-suite bathrooms and changes in lifestyle are contributing to the trend
towards using significantly more water for showering (Shaban and Sharma, 2007)

**CONCLUSION**
The increased demand for water has imposed a pressure on water supply system, which has led to
environmental problems such as over-exploitation of water resources and breaks in the balance of the
ecosystem. Determining the behavior of domestic water, consumers can facilitate a more proactive approach
to water demand management, and serves as the foundation for the development of any intervention
strategies that seek to bring about sustained and substantial reductions in domestic water consumption. This
paper presented the findings of a domestic water consumption questionnaire survey containing over
40 questions carried out in urban Batticaloa in Manmunai Pattu, Sri Lanka. Simple random sampling
technique was followed to select households and the statistical package IBM SPSS 25.0 was used for
data entry and analysis of the data.
This study showed that high income level as well as living standards increased total domestic water
consumption. It was shown that elder people use less water than younger people in general. Total



domestic water consumption for household uses indicated that, highest amount of (72%) water has
been used for showers and bath compared to toilet flushing, personal hygiene and cloth washing.
Family size and number of taps in a household were found to be important indicators in estimating
household water consumption; it was shown that families with many members and high number of
taps have higher water consumption in general. Results showed that the total domestic water
consumption is negatively correlated with education level. The findings of this study concluded that,
the socio-economic condition of the households impacts on various aspects of domestic water
consumption in urban Batticaloa in Manmunai Pattu, Sri Lanka.

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

Management, Philadelphia, PA, USA.