# Peer review of "\*Corresponding Author"

_Drinking Water Engineering and Science, 2020_

## Referee Comment (RC1) · Anonymous Referee #1 · 25 Oct 2020

The study is interesting. However, there are much doubt regarding the validity of the data collection. The statistical analyses conducted in this study are also simple. Why don't author use linear regression, for example? I strongly suggest authors do extra analysis, i.e. linear regression, to have stronger results.

The draft structure is also poor, e.g., start the sub-section by a table. I suggest accepting this draft ONLY if they can improve the manuscript significantly.

Abstract: Please state clearly what the benefits are in the abstract.

Keywords: change "efficiency" to "water efficiency".

Introduction: The household water consumption is determined by quite a few factors, such as climate, seasonality, socioeconomic characteristics, and socio-demographics

-> please give references.

Methods: 1. Please put the map of the location. 2. Please elaborate all statistical analyses you did in the methods section, e.g. correlation test, PCA, etc. 3. A Simple random sampling technique was followed to select households -> Please elaborate what you mean by "simple". 4. How did you come up with "75 households belonging to the 20 urban area in Batticaloa District in Manmunai Pattu"? 75 households are not enough to represent an urban area with, let say, 1000 households. if the total population is thousands, then 75 is too small. 5. How did you come up with the demographic variables used in the analysis? for example, why did you choose income and not other things? Please support with literature.

Results and discussion: 1. Never start a new section or sub-section by a table or picture. Always start by sentences/paragraph. Please edit the whole sub-section! 2. Please improve table 1. please see, for example, https://doi.org/10.1080/09603123.2016.1217314 (in table 2). 3. Please include mean/average, min, max in table 1, if applicable. 4. You already have table 1, so don't need to elaborate on the characteristics of the respondents in detail. 5. You don't need to show the table of Pearson correlation between 2 variables. write only the correlation value and p-value to shorten the draft. Please edit all tables with Pearson correlation! 6. Please improve all pie charts, they are in bad quality, maybe use a normal chart without variation; improve also chart in Figure 3. 7. There is no data that support all sentences in no. 2 (age of household members). This looks like empty discussion, without supporting data/results.

Conclusion: 1. Never write the software used in the conclusion! 2. What is/are the implication of this study??

---

## Referee Comment (RC2) · Anonymous Referee #2 · 3 Nov 2020

The authors have done a survey in a urban area in Sri Lanka which probably includes water use and household charactersitics. However, the survey questions were not included, so the reader does not know this. Also, doing a survey is an art, and it is not clear how skillfuly this art was performed in this study.

The method section is only 5 sentences long, and states that a survey was done. How is unclear.

The results section shows superfluous tables, with a lot of excess data; this could be much more compact. There are no linear regression results, no graphs either. There is no discussion on statistical significance of only 75 households being surveyed. There was no hypothesis on water use and its signifcant contributors (from e.g. literature on countries that are similar to Sri Lanka, how significant is USA data in this respect?)

and then a statisitcal test to (dis)prove the hypothesis.

Even if the data was approached in a scientific way, it still is no more than a case study. A nice set of data of water use in this specific area. There is no lesson to learn from this - there is no study on how water demand could be reduced, or something similar. The contribution of this study to the scientific community is not clear at all. The data was not even provided.

---

## Author Comment (AC1) · 11 Nov 2020

Reviewer 1 1. Why don't author use linear regression, for example? I strongly suggest authors do extra analysis, i.e. linear regression, to have stronger results. Models Based on Linear Regression

Table 2 R R Square Adjusted R Square Std. Error 0.982 0.965 0.962 23.8504

Linear regression analysis was conducted to examine the effects of the potential predictors on per capita water consumption in urban households. The model was statistically significant as household size, age, education level, number of taps and household income showed statistical significance ($p \leq 0.05$) and together they accounted for 96.5% of the variation in per capita water consumption in the wet season , $R^2$ = 0.965, F=

375.813. 2. The draft structure is also poor, e.g., start the sub-section by a table. I suggest accepting this draft ONLY if they can improve the manuscript significantly. Draft structure changed. 3. Abstract: Please state clearly what the benefits are in the abstract. Stated the benefit.

4. Keywords: change "efficiency" to "water efficiency". Changed 5. Methods: 1. Please put the map of the location. Included

6. Please elaborate all statistical analyses you did in the methods section, e.g. correlation test, PCA, etc. Statistical Analysis Quantitative data on socio-demographic and water consumption characteristics were entered into the Statistical Package for Social Sciences (SPSS 25.0). Correlation and linear regressions were used to examine the relationships between per capita water consumption and the potential predictors The predictors were chosen based on the review of the results of studies conducted by Thompson et al.,(2001) and House-Peters and Chang, (2011).. A linear regression analysis was used to determine the predictors of water consumption. Each variable was entered in a sequence and its value was assessed with statistical significance set at $p \leq 0.05$. 3. A Simple random sampling technique was followed to select households -> Please elaborate what you mean by "simple". Removed the word 4. How did you come up with "75 households belonging to the 20 urban area in Batticaloa District in Manmunai Pattu"? 75 households are not enough to represent an urban area with, let say, 1000 households. if the total population is thousands, then 75 is too small. A systematic random sampling was performed to select the sample households. The total number of households in the urban area was 300. A random number (K) was determined for the first selection. Desired sample size was estimated from the following equation. K = Population size / Sample size The estimated desired sample size was 75. Each household from the Manmunaipattu was given a unique identification number. From the list of households every 300th houses were selected and the data were collected. 5. How did you come up with the demographic variables used in the analysis? for example, why did you choose income and not other things? Please support

with literature. Cited under the statistical analysis sub section Results and discussion: 1. Never start a new section or sub-section by a table or picture. Always start by sentences/paragraph. Please edit the whole sub-section! Edited, Tables were replaced after the text.

2. Please include mean/average, min, max in table 1, if applicable. Variable Mean Max Min Std. Dev Age (Years) 51.41 72 31 11.3 Education (Years) 3.01 4 0 0.86 Average income (Rs/ month) 36,045.47 80,000 7,000 18,042.25 Family size 4.03 7 2 1.14 Number of taps 5.88 10 1 2.34 4. You don't need to show the table of Pearson correlation between 2 variables. write only the correlation value and p-value to shorten the draft. Please edit all tables with Pearson correlation! All the tables were edited Age of household members It is shown that the total domestic water consumption is negatively correlated with age and the correlation coefficient was -0.944 (p<0.01). Living standards Total domestic water consumption is positively correlated with living standards as p<0.01 (Table 2). Income level It is shown that the total domestic water consumption is positively correlated with income level and the correlation coefficient was 0.968 (p<0.01).

Education level The education level also influences the water consumption in a household. It is shown that the total domestic water consumption is negatively correlated with education level and the correlation coefficient was -0.873 (p<0.01) (Table 4). Number of taps The number of taps also influences the water consumption in a household. Table 5 shows that the total domestic water consumption is positively correlated with the number of taps and the correlation coefficient was 0.951 (p<0.01).

Household size Table 6 shows that the total domestic water consumption is positively correlated with household size and 0.95 (p<0.01).

5. Please improve all pie charts, they are in bad quality, maybe use a normal chart without variation; improve also chart in Figure 3.

6. There is no data that support all sentences in no. 2 (age of household members).

This looks like empty discussion, without supporting data/results. Supporting data was included.

Conclusion: 1. Never write the software used in the conclusion! Edited 2. What is/are the implication of this study?? These findings would be useful in managing the water demand and help to reduce the water consumption in urban areas.

Please also note the supplement to this comment:
https://dwes.copernicus.org/preprints/dwes-2020-32/dwes-2020-32-AC1-supplement.pdf

[Figure]

**STUDY AREA**

SRI LANKA

BATTICALOA DISTRICT

0   3.75  7.5      15      22.5      30
Miles

**Fig. 1.** Map

---

## Author Comment (AC2) · 11 Nov 2020

Reviewer 2 1. The authors have done a survey in a urban area in Sri Lanka which probably includes water use and household characteristics. However, the survey questions were not included, so the reader does not know this. Survey questions were attached.

2. Also, doing a survey is an art, and it is not clear how skillfuly this art was performed in this study. The method section is only 5 sentences long, and states that a survey was done. How is unclear. Methodology was elaborated

3. The results section shows superfluous tables, with a lot of excess data; this could be much more compact. Removed excess data

4. There are no linear regression results, no graphs either. Linear regression results

were included.

5. There is no discussion on statistical significance of only 75 households being surveyed. The method was explained.

6. There was no hypothesis on water use and its signifcant contributors (from e.g. literature on countries that are similar to Sri Lanka, how significant is USA data in this respect?)and then a statisitcal test to (dis)prove the hypothesis. H0 = There is no relationship between the dependent variable percapita water consumption and the independent variables; household size, age, education level, number of taps and household income. Ha= There is a relationship between the dependent variable percapita water consumption and the independent variables; household size, age, education level, number of taps and household income. 7. Even if the data was approached in a scientific way, it still is no more than a case study. A nice set of data of water use in this specific area. There is no lesson to learn from this - there is no study on how water demand could be reduced, or something similar. The contribution of this study to the scientific community is not clear at all. The data was not even provided.

Please also note the supplement to this comment:
https://dwes.copernicus.org/preprints/dwes-2020-32/dwes-2020-32-AC2-supplement.pdf

---

## Author Comment (AC3) · 11 Nov 2020

The comment was uploaded in the form of a supplement:
https://dwes.copernicus.org/preprints/dwes-2020-32/dwes-2020-32-AC3-supplement.pdf

---

## Author Comment (AC4) · 11 Nov 2020

**QUESTIONNAIRE**

**Domestic water consumption pattern in urban Batticaloa, Manmunai Pattu**

1) **General**

   1.1 G.N Division     : …………………….
   1.2 Town             : …………………….
   1.3 Ethnicity        : …………………….
   1.4 Religion: …………………….

2) **Respondent**

   2.1 Name    : ………………………….
   2.2 Sex       : (1) Male (2) Female

3) **Household information**

   3.1 Name of the household head     : …………………….
   3.2 Family size                    : ………
   3.3 Average monthly total household income (Rs.)    : ………
      (1) 5,000 – 10,000                          (5) 25,001 – 30,000
      (2) 10,001 – 15,000                        (6) 30,001 – 40,000
      (3) 15,001 – 20,000                        (7) 40,001 – 50,000
      (4) 20,001 – 25,000                        (8) > 50,000
   3.5 Living standard of the family (Interviewer point of view)

      (1) Poor                    (2) Medium                    (3) Rich

   3.6 House ownership

      (1) Own
      (2) Shared
      (3) Rented
      (4) Temporary in own land
      (5) Others

   3.7 Family details

| No | Relationship to HH Head | Age | Sex | Marital status | Education | Occupation | Income |
|----|------------------------|-----|-----|----------------|-----------|------------|--------|
|    |                        |     |     |                |           |            |        |
|    |                        |     |     |                |           |            |        |
|    |                        |     |     |                |           |            |        |

| | | | | | | | |
|---|---|---|---|---|---|---|---|
| | | | | | | | |
| | | | | | | | |
| | | | | | | | |
| | | | | | | | |
| | | | | | | | |
| | | | | | | | |

- Relationship: (1) Husband (2) Wife (3) Son (4) Daughter (5) Parents (6) Grandparent (8) Siblings (9) Others
- Marital status: (1) Single (2) Married (3) Divorced (4) Separated (5) Windowed
- Occupation: (1) Government (2) Private/NGO (3) Business (4) Farmer (5) Fishing (6) Day-wage labour (7) Household work (8) Others
- Education: (1) Primary (1-5) (2) Intermediate (6-11) (3) Advanced (12-13) (4) higher (dip/graduate) (5) Non
- Income (Rs.)

| | | | |
|---|---|---|---|
| (1) 5,000 – 10,000 | (3) 15,001 – 20,000 | (5) 25,001 – 30,000 | (7) 40,001 – 50,000 |
| (2) 10,001 – 15,000 | (4) 20,001 – 25,000 | (6) 30,001 – 40,000 | (8) > 50,000 |

**4. Household Water Consumption of water per day**

4.1 What is the major source of household water supply?

1. Pipeline water
2. Dug Well
3. Tube well
4. Other

4.2 What are the major water using appliances in your home?

1. Shower
2. Flushing toilet
3. Hand basin
4. Bath tub
5. Washing machine
6. Water heater
7. Other: Specify

4.3 How many taps are there in the household?

1. One
2. Two
3. Three
4. Four
5. Five and more

4.4 Consumption of water per day (Common)

| Water Use | Consumption of water per day | | | | | | | | | | | | | | | | | | | | | | | | | | | | | | | |
| --- | --- | --- | --- | --- | --- | --- | --- | --- | --- | --- | --- | --- | --- | --- | --- | --- | --- | --- | --- | --- | --- | --- | --- | --- | --- | --- | --- | --- | --- | --- | --- | --- |
| | Pipeline | | | | | | | | Well | | | | | | | | Tube well | | | | | | | | Others | | | | | | | |
| | Age group | | | | | | | | Age group | | | | | | | | Age group | | | | | | | | Age group | | | | | | | |
| | 0-4 | 5-14 | 15-24 | 25-34 | 35-44 | 45-54 | 55-64 | 64< | 0-4 | 5-14 | 15-24 | 25-34 | 35-44 | 45-54 | 55-64 | 64< | 0-4 | 5-14 | 15-24 | 25-34 | 35-44 | 45-54 | 55-64 | 0-4 | 5-14 | 15-24 | 25-34 | 35-44 | 45-54 | 55-64 | 64< |
| Toilets | | | | | | | | | | | | | | | | | | | | | | | | | | | | | | | | |
| Bathing | | | | | | | | | | | | | | | | | | | | | | | | | | | | | | | | |
| Drinking | | | | | | | | | | | | | | | | | | | | | | | | | | | | | | | | |
| **Total** | | | | | | | | | | | | | | | | | | | | | | | | | | | | | | | | |

4.5 Consumption of water per day (Specific)

| Water Use | Consumption of water per day | | | |
| --- | --- | --- | --- | --- |
| | **Pipeline** | **Well** | **Tube well** | **Others** |
| Clothes washing | | | | |
| Utensils Cleaning | | | | |
| Cooking | | | | |
| Watering the Garden | | | | |
| House Cleaning | | | | |
| Others | | | | |
| **Total** | | | | |

**5. Water use habits**

**5.1 Bathing habits**
5.1.1 Your usual way of washing yourself is:

1. Shower                    2. Bath                    3. Rubbing with towel

5.1.2 Frequency of shower is:

1. More than 2 times/day      3. Once every day
2. 2 times/day                4. Once every 2 days        5. Longer

5.1.3 Time length of each shower:

1. Less than 5 min            3. 10 min                   5. More than 30 min
2. 5 min                      4. 20 min

**5.2  Utensils cleaning habits**
5.2.1 Do you wash dishes at home?

1. Yes                                        2. No

5.2.2 Frequency of washing:

1. More than 2 times/day      3. Once every day           5. Longer
2. 2 times/day                4. Once every 2 days

5.2.3 Time length of each wash:

1. Less  than                 2. 10 min                   4. >30 min
   10 min                     3. 30 min

**5.3 Drinking water habits**
5.3.1 What is the major source of household drinking water?

1. Piped water               3. Bottle water             5. Other: Specify

2. Well water                4. Tube well

5.3.2 What is the general family practice adopted in drinking water?

1. Boil and Drink
2. Filter and Drink
3. Drink without boiling or filtering
4. Combination of method

**5.4 Outdoor use**

How much of your lot area is watered (irrigated)

During a typical dry season, how frequently do you irrigate?

1. Less  than  once  a        2. Once a week              4. Daily
   week                       3. Every other day

When do you irrigate?

1. Early morning        3. Afternoon
2. Late morning         4. Evening

How do you irrigate? (Please check all that apply)

1. By hand (hose or bucket)
2. Manual sprinkler (one you move around)
3. In-ground sprinkler
4. Other (please specify

Do you use any additional sources for irrigation water? (Please check all that apply)

1. No
2. Nearby surface water (stream, pond, river, lake)
3. Rain barrel
4. Purchase water

How were you affected by last year's drought?

1. No problem
2. Not enough water to irrigate as much as I wanted to
3. Couldn't irrigate at all
4. Well(s) went completely dry

**6. Water supply**

5.1 Do you know how much do you pay for each cubic meter of water?

1. Yes
2. No
3. Other: Specify

5.2 What do you think about the current water rate?

1. Too high
2. Normal
3. Too low
4. Do not know

5.3 Do you face an irregular water supply?

1. Yes
2. No

5.4 Quality of your domestic tap water

1. Turbidity (1) Transparent (2) Slightly turbid (3) Moderately turbid (4) severely turbid
2. Color (1) Colorless (2) Others__________
3. Taste (1) Good (2) Just so (3) Uncomfortable/Bad
4. Smell (1) Normal (2) Abnormal

5.5 Quality of your supply water

1. Turbidity (1) Transparent (2) Slightly turbid (3) Moderately turbid (4) severely turbid
2. Color (1) Colorless (2) Others__________
3. Taste (1) Good (2) Just so (3) Uncomfortable/Bad
4. Smell (1) Normal (2) Abnormal

**7. Water awareness**

7.1 Do you limit how much water you use for any of these reasons?

1. Not sure well has enough water
2. Keep electrical bill down
3. Keep water bill down
4. Not sure septic system can handle all wastewater
5. Want to conserve water to protect the resource
6. Other (Please specify)

7.2 Have you done any of these actions to conserve water?

1. Take shorter showers
2. Installed low-flow plumbing fixture(s)
3. Water outdoors during early morning or evening
4. Installed a water efficient irrigation system
5. Reduced landscape area irrigated
6. Other (Please specify)

7.3 How do you deal with running or leaky toilets and other faucets?

1. Never had the problem
2. Repair running toilet immediately
3. Call a plumber immediately
4. Fix leaks within one week
5. Fix leaks eventually
6. Other (Please specify)

7.4 Are you concerned about the quality of your water?

1. No
2. Yes, we drink only bottled water
3. Yes, we have had our well water tested during the past year
4. Yes, we look at the water quality report sent by our water company
5. Yes, we have our own treatment system
6. Other (Please specify)